# *Actinidia eriantha Benth*. Root as a New Phytomedicine Inhibits Non-Small Cell Lung Cancer by Regulating·TGF-β/FOXO/mTOR

**DOI:** 10.3390/ijms26188957

**Published:** 2025-09-14

**Authors:** Xuan Zhang, Qiyao Xiao, Haoran Chen, Shaoming Yang, Qingli Li, Lihua Peng

**Affiliations:** 1College of Pharmaceutical Sciences, Zhejiang University, Hangzhou 310058, China; zhangxuanwzj@163.com (X.Z.); 22219104@zju.edu.cn (Q.X.); 22219130@zju.edu.cn (H.C.); 2Jinhua Institute, Zhejiang University, Jinhua 321299, China; 3Longquan Industrial Innovation Research Institute, Zhejiang University, Longquan 323700, China; 13666565365@163.com (S.Y.); liqingli888@126.com (Q.L.)

**Keywords:** *Actinidia eriantha Benth*. root, extract, non-small cell lung cancer, TGF-β/FOXO/mTOR pathway

## Abstract

Non-small cell lung cancer (NSCLC) accounts for over 80% of lung cancer cases and remains challenging to treat due to high recurrence and mortality rates. While cisplatin (CDDP) is a first-line chemotherapy for NSCLC, its clinical utility is limited by toxicity, drug resistance, and inadequate tumor suppression. Plant-derived extracts have shown promise as complementary therapies, offering potential benefits including enhanced efficacy and reduced treatment side effects. The aqueous extract of *Actinidia eriantha Benth*. root (WE-AER) exhibits significant antitumor activity, though its mechanisms in NSCLC remain unclear. This study demonstrates WE-AER’s potent anti-NSCLC effects through multiple mechanisms. In vitro, WE-AER dose-dependently inhibited A549 cell growth (74.74% inhibition, IC50 = 210.38 µg/mL) by inducing apoptosis via TGF-β/FOXO/mTOR pathway modulation. In vivo, WE-AER suppressed tumor growth and angiogenesis while activating immune responses and reducing inflammation in mouse models, with excellent biosafety. These findings elucidate WE-AER’s anticancer mechanisms and support its potential as a novel herbal therapy for NSCLC.

## 1. Introduction

Lung cancer is one of the most prevalent and deadly malignancies, with the highest mortality rate globally [1,2]. Non-small cell lung cancer (NSCLC) is the most common form, and its primary subtypes include lung adenocarcinoma (LUAD), lung squamous cell carcinoma (LUSC) and large cell carcinoma [3,4]. The primary treatment options for NSCLC include surgery, chemotherapy, and radiotherapy. Notably, platinum-based chemotherapy, such as cisplatin (CDDP), is the standard first-line treatment for NSCLC. Nevertheless, this approach is often limited by significant challenges, including severe side effects, drug resistance, and high costs. Additionally, prolonged use of these drugs can lead to reduced therapeutic efficacy and potential long-term damage to the body [5,6,7,8,9]. Therefore, developing effective, low-toxicity anticancer agents is crucial for NSCLC treatment [10,11].

Traditional Chinese medicine (TCM), with its extensive history in treating malignant tumors, has garnered growing interest. It provides a unique approach that is noted for its minimal side effects, long-lasting therapeutic outcomes, and its holistic ability to regulate the body [6,12,13,14,15]. *Actinidia eriantha Benth* (AE) is a liana plant typically found in temperate regions, and is a traditional medicinal herb used by the She ethnic minority in China. Its roots (AER) have been traditionally used in TCM to treat illnesses including gastric, nasopharyngeal, and breast carcinomas, as well as hepatitis [16]. Pharmacological studies have demonstrated that the aqueous extracts of AER exhibit antitumor and immune-boosting properties [17]. Additionally, several triterpenoid compounds, such as ursolic acid, eriatic acid A, and eriantic acid B, have been isolated from AER [17,18,19,20]. However, the full spectrum of active chemical components in the extract remains unclear. Despite its widespread use in TCM for treating malignant tumors, there is a lack of modern scientific evidence to support its efficacy against lung cancer. Furthermore, the anticancer mechanisms and safety profile of AER extract have yet to be thoroughly investigated.

Based on these traditional uses and previous pharmacological findings, we hypothesized that the aqueous extract of *A. eriantha* root (WE-AER) may exert antitumor effects against NSCLC by modulating key signaling pathways involved in cell proliferation and apoptosis. The objectives of this study were to demonstrate the anti-NSCLC efficacy of WE-AER both in vivo and in vitro and to uncover the mechanisms underlying its anticancer effects through transcriptome analysis and molecular functional analysis, thereby evaluating its potential as a promising therapeutic candidate for treating NSCLC.

## 2. Results

### 2.1. Characterization and Bioactivities Screening of the ARE Extracts

The effects of AER on the proliferation of lung cancer cell lines are illustrated in Figure 1a. As shown in Figure 1b, 400 μg/mL of WE-AER significantly reduced the cell viability of A549 cells, achieving an inhibition rate of 74.74%. The half-maximal inhibitory concentration (IC50) of WE-AER for A549 cells was 210.38 μg/mL. In contrast, alcoholic extract of AER (EE-AER) showed no significant growth inhibitory effect on A549 cells (*p* > 0.05) (Figure 1c). Notably, WE-AER displayed the strongest growth inhibitory effect on A549 cells (*p* < 0.01). Thus, in subsequent experiments, we focused primarily on WE-AER and human non-small lung cancer A549 cells to further investigate the underlying mechanisms of WE-AER against A549.

Furthermore, LCMS/MS analysis was performed to analyze the small molecular components in WE-AER. The results revealed that 16,311 small molecules were identified, with unnamed compounds being predominant (95.67%) (Figure 1d). Based on class classification, the top three named compounds were organic acids and their derivatives, lipids and lipid-like molecules, and organic heterocyclic compounds (Figure 1e). Meanwhile, we conducted comprehensive structural characterization of representative small molecules in WE-AER, with the top-ranked compounds (top ten) highlighted in blue (Figure 1f). Additionally, further information on the small molecules was provided in Appendix A. Subsequently, we identified bioactive small-molecule categories with potential anticancer properties in WE-AER and elucidated their mechanisms of action, primarily including apoptosis induction, anti-migration, anti-inflammatory effects, inhibition of DNA fragmentation within tumors, and suppression of DNA damage repair (Figure 1g). Notably, L-(+)-linoleic acid (62.38%), creatinine (7.89%), and linolenic acid (6.62%) were identified as the three most abundant small molecules in WE-AER (Figure 1h). Interestingly, L-(+)-linoleic acid and linolenic acid have been individually reported in previous studies to possess potential antitumor properties [21,22]. These findings suggest that WE-AER contains a mixture of bioactive constituents, which may collectively contribute to its overall antitumor activity.

### 2.2. Effects of WE-AER on the Migration, Invasion Inhibition, and Pro-Apoptosis of A549 Cells

Cell migration and invasion assays are widely used to evaluate tumor malignancy phenotypes [23]. To assess the impact of WE-AER on A549 cell migration and invasion, a Transwell assay was conducted. The results revealed that WE-AER significantly inhibited migration and invasion of A549 cells after 24 h of intervention compared to the control group. Notably, the maximum inhibition rates for migration and invasion reached 60% and 39%, respectively (Figure 2a–d). Additionally, based on earlier pathway predictions, the pro-apoptotic effects of WE-AER on A549 cells were examined. Flow cytometry analysis indicated that the apoptosis rate of A549 cells increased to nearly 30% after 24 h of WE-AER treatment, which was approximately three times higher than the control (Figure 2e,f). These results indicate that WE-AER treatment is associated with inhibited migration and invasion as well as promoted apoptosis in A549 cells under the tested conditions.

### 2.3. Anticancer Mechanism of WE-AER

To investigate the potential pathways through which WE-AER effects A549 cells, high-throughput sequencing technologies were performed (Figure 3). Principal component analysis (PCA) was performed based on gene expression levels to assess sample repeatability and identify potential outliers. The Pearson correlation coefficient between samples further confirmed high intra-group reproducibility but weak inter-group correlation (Figure 3a). Additionally, 9304 genes were commonly expressed in both groups (Figure 3b). A total of 6980 differential genes were identified, including 1000 up-regulated genes and 5980 down-regulated genes, indicating that WE-AER significantly down-regulated A549-related genes expression (Figure 3c). Further analysis of the KEGG and GO results for the top 20 differentially expressed genes revealed significant enrichment in cancer-related pathways, particularly the TGF-β, Foxo, and mTOR signaling pathways (Figure 3d–f), all of which play key roles in cancer cell genesis, development, apoptosis, and metabolism [24,25,26]. WE-AER was shown to exert its anti-A549 effect by inhibiting the expression of the proteins associated with these major pathways, as demonstrated by Western blot analysis (Figure 3(g1–i2)). Meanwhile, these results indicated that WE-AER exerted anti-A549 effects by down-regulating the expression of TGF-β, Foxo, and mTOR signaling pathways (Figure 3j).

### 2.4. Anticancer Effects of WE-AER in Nude Mice with Xenograft Tumors

In this study, a xenograft model was established in BALB/c nude mice to evaluate the in vivo effects of WE-AER on tumor growth (Figure 4a). As observed in Figure 4b, tumor development was observed at 21 d post-treatment in both the treatment and control groups. Figure 4c shows that tumors in the control group grew rapidly over the 21-day period, reaching an average size of about 700 mm^3^. However, the administration of the positive control (PTX) and WE-AER significantly slowed down the tumor growth trend. Notably, the tumor-suppressing effect of WE-AER was found to be comparable to that of PTX, a standard chemotherapy agent. In addition, the average tumor weight in the control group was 400 mg, which was roughly four times that of the treatment group (Figure 4d). As seen in Table 1, the effects of each group on organ indexes of mice were summarized. Compared with the control group, the PTX group showed a decrease in liver and kidney indices, and there was no significant difference in the WE-AER group, which may be related to the obvious liver and kidney toxicity of PTX [27].

H&E staining, CD31, VEGF, and TUNEL assays were conducted to elucidate the antitumor effects of WE-AER in vivo. As illustrated in Figure 4e–h, immunofluorescence results from the H&E staining images, and TUNEL, CD31, and VEGF staining were presented. Firstly, the H&E staining images revealed that the tumor tissues in the control group showed high-density tumor cells arranged in a disordered and unorganized manner. The cells had overlapping nuclei, with little to no necrosis observed [28,29]. In contrast, following treatment with PTX and WE-AER, the number of tumor cells was significantly reduced. Additionally, the tumor cell nuclei showed lighter staining and gradual cell structure disappearance, along with noticeable vacuolization. Subsequently, the immunofluorescence results of TUNEL, CD31, and VEGF showed that compared to the control group, PTX and WE-AER treatments significantly enhanced apoptosis of A549 cells. Moreover, both treatments reduced the levels of VEGF and CD31 in the tumor cells, highlighting their ability to inhibit angiogenesis in vivo, as confirmed by previous studies [30,31]. In Figure 5d–g, the effect of drug administration on the level of inflammatory factors in serum was evaluated. Compared with the control group, the levels of proinflammatory factors IL-1β and IL-6 in serum were decreased, and the levels of anti-inflammatory factor IL-10 and tumor necrosis factor TNF-α were increased in the WE-AER group, and there was no significant difference between the WE-AER group and PTX group [32,33]. These results suggested that WE-AER anti-A549 in vivo functions by modulating immunity and alleviating the inflammatory response in mice.

In addition, TGF-β, Foxo, and mTOR signaling pathways have been widely demonstrated to play pivotal roles in the regulation of cell growth, differentiation, and development. It also importantly contributes to the formation, development, metastasis, and neovascularization of malignant tumor cells [34,35,36,37]. In vitro, we have shown that WE-AER significantly regulates several proteins in the TGF-β, Foxo, and mTOR signaling pathways. Meanwhile, in vivo, we confirmed that the treatment of A549 mice with WE-AER was dependent on the TGF-β, Foxo, and mTOR signaling pathways using Western blot analysis. As shown in Figure 5(a1–c1), the expression levels of key proteins associated with these pathways were markedly reduced following treatment. This reduction in protein expression was accompanied by the inhibition of A549 tumor growth and angiogenesis. These findings provide robust evidence supporting the anticancer efficacy of WE-AER in vivo, highlighting its potential therapeutic impact through the modulation of critical signaling pathways involved in tumor progression.

### 2.5. The WE-AER with High Biosafety In Vivo

Figure 6a,b shows that the weight and survival curves of the mice changed in each group over a 21-day period. Compared to the control group, there was no significant difference observed in the WE-AER group, except for a decrease in body weight in the PTX group and the onset of mouse deaths from day 15 onwards. Additionally, as shown in Figure 6c, the hepatocyte distribution in the PTX group exhibited slight disorganization and pathological changes, while renal cells showed a slightly disordered arrangement. However, the deterioration was not substantial, which is consistent with previous findings [38]. It is noteworthy that other H&E tissue sections showed normal findings. In addition, H&E tissue sections of heart, liver, spleen, lung, and kidney in the WE-AER group showed no visible pathological changes, further supporting the excellent biocompatibility of WE-AER in antitumor applications in vivo.

## 3. Discussion

*Actinidia eriantha Benth.* root (AER) has been traditionally used in Chinese folk medicine for the treatment of various cancers, including gastric carcinoma, nasopharyngeal carcinoma, and breast carcinoma [17]. However, its anticancer activity against A549 remains underexplored. Meanwhile, there is limited literature on the mechanistic analysis of this plant’s effects. In this study, we provide a comprehensive mechanistic investigation demonstrating that the aqueous extract of AER (WE-AER) suppresses the proliferation of A549 cells by modulating the TGF-β/FOXO/mTOR signaling pathway, subsequently inducing apoptosis and inhibiting cell migration and invasion. Most importantly, WE-AER significantly inhibited tumor growth in A549 cell-derived xenograft models in nude mice, highlighting its potent anticancer activity against lung cancer(Figure 7).

Unrestrained cellular proliferation, impaired apoptosis, dysregulated cell cycle progression, and heightened migratory capacity are hallmark characteristics of tumor cells, making them critical targets for therapeutic intervention [39,40,41,42]. Our findings indicated that WE-AER induced dose-dependent growth suppression in A549 cells (Figure 1). Furthermore, it effectively inhibited the migration, invasion, and pro-apoptotic capabilities of A549 cells. Additionally, we demonstrated the optimal anticancer efficacy of WE-AER in suppressing A549 tumor growth in nude mouse models. Nude mice serve as an ideal model for evaluating human cancer xenografts and chemotherapy efficacy, primarily due to their congenital T-cell deficiency, which creates an immunocompromised environment conducive to tumor growth. LCMS/MS analysis revealed that WE-AER is a complex mixture, with linoleic acid, creatinine, and linolenic acid constituting the most abundant small molecules. While these specific compounds have been associated with antitumor effects in other model systems [21,22], it is crucial to emphasize that the pronounced anticancer efficacy of WE-AER observed in our study is likely the result of synergistic interactions between multiple constituents within the whole extract, rather than the effect of any single compound. This holistic effect is a common characteristic of phytomedicines and underscores the advantage of using a multi-component extract over isolated single entities. Future studies to fractionate the extract and evaluate the activity of individual compounds will be essential to identify the most pharmacologically active components and clarify their specific roles and potential synergies.

Preliminary investigation experiments showed that WE-AER was effective against A549 and exhibited good biocompatibility when compared to the chemotherapeutic agent PTX (Figure 6). Transcriptomic analysis and molecular validation revealed that WE-AER exerts its effects primarily through modulation of the TGF-β/FOXO/mTOR signaling pathways (Figure 3), which play pivotal roles in the regulation of cell growth, differentiation, development, and contribute significantly to the formation, development, metastasis, and neovascularization of malignant tumor cells [34,35,36,37]. In terms of apoptosis, WE-AER was effective in promoting apoptosis in lung cancer cells by enhancing the intensity of tunneling fluorescence. In addition, compared with PTX, WE-AER effectively reduced the production of inflammatory factors (IL-1β and IL-6) and enhanced the production of anti-inflammatory factors (IL-10) and tumor necrosis factors (TNF-α). It also effectively inhibited the expression of vascular endothelial growth factor CD31 and epidermal growth factor receptor [43]. Therefore, WE-AER inhibited the expression of inflammatory factors and tumor neoendothelial factors to a large extent, regulated the immune response, changed the tumor microenvironment, promoted cell apoptosis, and further exerted anti-A549 efficacy.

Beyond the molecular mechanisms, our in vivo findings demonstrated that WE-AER treatment significantly suppressed tumor growth and angiogenesis while modulating immune responses in the tumor microenvironment. The reduction in VEGF and CD31 expression, coupled with the modulation of inflammatory cytokines, suggests that WE-AER creates an unfavorable environment for tumor growth and metastasis. These multi-faceted effects position WE-AER as a promising multi-targeted therapeutic agent for NSCLC.

Taken together, these results demonstrated that WE-AER had significant apoptosis-inducing activity in A549 cells, and it also exhibited anti-angiogenic effects, including growth inhibition, apoptosis induction, and inhibition of migration and invasion in A549 cells. Additionally, WE-AER suppressed angiogenesis in tumor tissues of nude mice with xenograft tumors. Meanwhile, WE-AER possessed significant anti-inflammatory, immunomodulatory, and tumor microenvironment improvement efficacy. In addition, findings further revealed that WE-AER enhances its anti-A549 efficacy by interfering with the expression of key signaling proteins related to the TGF-β/Foxo/mTOR pathway, which are crucial for regulating cell growth, survival, and apoptosis. Importantly, WE-AER demonstrated high biocompatibility, which is crucial for its potential therapeutic use in NSCLC treatment. These results provide strong supporting evidence for the application of WE-AER as a promising treatment option for NSCLC.

## 4. Materials and Methods

### 4.1. Chemical Reagents and Materials

*Actinidia eriantha Benth*. root was provided by Zhejiang Longquan Zhengda Biological Technology Co., Ltd. (Lishui, China). Human non-small lung cancer A549 cells were purchased by Wuhan Pricella Biotechnology Co., Ltd. (Wuhan, China) F-12K medium, trypsin and penicillin–streptomycin solution were purchased from HyClone, Logan, UT, USA. The fetal bovine serum (FBS) was from Hyclone Co. (Fremont, CA, USA). Cell count kit-8 (CCK-8), Triquick reagent (Trizol substitute), 5% BSA blocking buffer, Triton X-100, RIPA buffer, and ECL Western blotting substrate were purchased from Solarbio, Beijing, China. Annexin V-fluorescein isothiocyanate (FITC)/propidiumiodide (PI) apoptosis detection kit was purchased from Nanjing Key GenBiotech CO., Ltd. (Nanjing, China). Chloroform, acetonitrile, n-butanol, Tris-HCl, and Sodium dodecyl sulfate (SDS) were purchased from Sinopharm Chemical Regent Co., Ltd., Shanghai, China. BCA protein assay kit was purchased from Biosharp, Hefei, China. PVDF transfer membrane for Western blotting was purchased from Merck Millipore, Billerica, MA, USA. Mouse IL-1β, IL-6, IL-10 and TNF-α ELISA Kits were purchased from Boster, San Mateo, CA, USA. The columns for UHPLC were purchased from Thermo Fisher Scientific, Waltham, MA, USA. Matrigel and Transwell plates (8 μm, 3422) were purchased from Corning Co., Ltd. (Corning, NY, USA).

### 4.2. Extraction and Purification of Extracts from AER

Initially, the AER was soaked in double-distilled water or anhydrous ethanol (1:10, *w*:*v*) and boiled under reduced pressure at 100 °C three times, followed by micro-boiling for 1 h to obtain the crude aqueous or ethanol extract slurry. Both the aqueous and ethanol extracts were then filtered through a büshi funnel under vacuum filtration. The filtrate was concentrated in a rotary evaporator (RE-52A, Shanghai Yarong Biochemical Instrument Factory, Shanghai, China) under reduced pressure at 55 °C, followed by centrifugation at 3000 rpm for 15 min. The supernatant was precipitated with three volumes of anhydrous ethanol or double-distilled water, and stored overnight at 4 °C. The resulting supernatant was then concentrated to one-third of its original volume under reduced pressure to obtain the aqueous or ethanol extracts of AER. Finally, the supernatant was dried at 55 °C and stored in a sealed container to yield the aqueous (WE-AER) or alcoholic (EE-AER) extract of AER. Liquid chromatography–tandem mass spectrometry (LCMS/MS) was employed to analyze the components in WE-AER. LC-MS/MS analysis was performed under the following conditions:

Analysis was performed using a UHPLC (1290 Infinity LC, Agilent Technologies, Santa Clara, CA, USA) coupled to a quadrupole time-of-flight (AB Sciex TripleTOF 6600) in Shanghai Applied Protein Technology Co., Ltd. (Shanghai, China). For HILIC separation, samples were analyzed using a 2.1 mm × 100 mm ACQUIY UPLC BEH 1.7 µm column (Waters, Drinagh, Ireland). In both ESI-positive and -negative modes, the mobile phase contained A = 25mM ammoniumacetate and 25 mM ammoniumhydroxide in water and B= acetonitrile. The gradient was 85%B for 1 min and was linearly reduced to 65% in 11 min, and then was reduced to 40% in 0.1 min and kept for 4 min, and then increased to 85% in 0.1 min, with a 5 min re-equilibration period employed.

The ESI source conditions were set as follows: Ion Source Gas1 (Gas1) as 60, Ion Source Gas2 (Gas2) as 60, curtain gas (CUR) as 30, source temperature: 600 °C, and IonSpray Voltage Floating (ISVF) ± 5500 V. In MSonly acquisition, the instrument was set to acquire over the m/z range 60–1000 Da, and the accumulation time for TOF MS scan was set at 0.20 s/spectra. In auto MS/MS acquisition, the instrument was set to acquire over the m/z range 25–1000 Da, and the accumulation time for product ion scan was set at 0.05 s/spectra. The product ion scan is acquired using information-dependent acquisition (IDA) with high sensitivity mode selected. The parameters were set as follows: the collision energy (CE) was fixed at 35 V with ± 15 eV; declustering potential (DP), 60 V (+) and −60 V (−); exclude isotopes within 4 Da, candidate ions to monitor per cycle: 10.

### 4.3. Cell Viability Assay

A549 cells were grown in F-12K medium containing 10% FBS and 1% penicillin–streptomycin under standard culture conditions (37 °C, 5% CO_2_). The inhibitory effects of WE-AER and EE-AER on cell growth were assessed via the CCK-8 assay. Briefly, 5 × 10^3^ cells were seeded in 96-well plates, and various concentrations of WE-AER and EE-AER (0, 25, 50, 100, 200, and 400 μg/mL) were added after cell attachment. Following a 24 h incubation period, the absorbance at 450 nm was measured using a microplate reader (Multiskan FC, Thermo Fisher, Waltham, MA, USA).

### 4.4. Cell Migration and Invasion Assays

Cell migration and invasion were evaluated using Transwell assays, with Matrigel (dilution 1:8, 0827045, ABW) applied to the upper chamber for invasion measurements and omitted for migration measurements. The plate was allowed to stand for 30 min at 37 °C in the culture environment. After the matrix gel solidified, A549 cells (5 × 10^4^ per well) were seeded in basal medium, and the lower chamber contained 10% FBS medium. In the intervention group, 400 μg/mL WE-AER was added. After incubation for 24 h, cells on the upper membrane were removed, and those on the lower membrane were fixed, stained, and imaged using a microscope (ECLIPSE-TI-S, Nikon, Tokyo, Japan).

### 4.5. Cell Apoptosis Assay

A549 cells were inoculated into 6-well plates at a density of 1 × 10^5^ cells/well. After achieving 80% confluence, the cells were treated with 400 μg/mL WE-AER for 24 h. Subsequently, the cells were collected, stained, filtered through a cell sieve, and evaluated for apoptosis positivity using flow cytometry (CytoFlex S, Beckman, Brea, CA, USA).

### 4.6. Molecular Mechanism Analysis

High-throughput sequencing technologies were employed to further investigate the molecular mechanism of WE-AER against A549 cells. Briefly, rRNAs were removed using rRNA depletion kits, and the purified mRNAs were fragmented and reverse-transcribed into cDNAs. The cDNAs were then processed through end-pairing, A-tailing, and adapter ligation. The target fragments were separated via agarose gel electrophoresis, and the effective concentration of the library was determined by real-time reverse transcription polymerase chain reaction (RT-qPCR). The sequences of the primers used were as follows: Index Primer for Illumina: 5′-AATGATACGGCGACCACCGAGATCTACAC[i5Index]ACACTCTTTCCCTACACGACGCTCTTCCGATCT-3′. Index Primer for Illumina: 5′-CAAGCAGAAGACGGCATACGAGAT[i7Index]GTGACTGGAGTTCAGACGTGTGCTCTTCCGATC-3′. High-throughput sequencing was performed with Illumina HiSep and MiSep platforms. After processing the raw reads, the clean reads were aligned with reference transcripts using HISAT2 and Bowtie2 software to calculate fragments per kilobase of per million mapped fragments (FPKM) values, followed by differential gene expression analysis. Differential expressed genes were identified using the R package limma (version 3.60.0), applying the criteria of |log2FC| ≥ 1 and FDR < 0.05. These genes were then subjected to Gene Ontology (GO) functional enrichment and Kyoto Encyclopedia of Genes and Genomes (KEGG) pathway analysis. Results were considered statistically significant at *p* < 0.05.

### 4.7. Western Blot Analysis

Total protein was isolated using RIPA buffer supplemented with a protease inhibitor mixture and quantified using a BCA protein quantification kit. Gels consisting of 8–12% separating gel and 5% stacking gel were prepared, and 60 μg of total protein (10–15 μL) per well was loaded and separated at 60 V for the stacking gel and 80 V for the separating gel for 2 h. For protein transfer, PVDF membranes were soaked in methanol and then equilibrated in Tris-glycine transfer buffer containing 5% methanol. After transfer, the membranes were blocked with T-TBS containing 5% BSA for 1 h at room temperature, followed by incubation with primary antibody against p-PI3K (ab182651 Abcam, dilution 1:500), PI3K (ab133595 Abcam, dilution 1:2000), p-Akt (4060 CST, dilution 1:2000), Akt (4685 CST, dilution 1:1000), p-mTOR (5536 CST, dilution 1:1000), mTOR (2983 CST, dilution 1:1000), p-FoxO3(ab154786 Abcam, dilution 1:2000), FoxO3 (2497 CST, dilution 1:1000), TGF-β1 (ab179695 Abcam, dilution 1:1000), p-Smad2 (18338 CST, dilution 1:1000), Smad2 (5339 CST, dilution 1:1000), p-Smad3 (9520 CST, dilution 1:1000), Smad3 (9523 CST, dilution 1:1000), and β-actin (ab8226 Abcam, dilution 1:2000) at 4 °C overnight. The membranes were washed three times with T-TBS, incubated with Goat anti-Mouse (IgGH + L) (31160 Thermo Pierce, dilution 1:5000) and Goat anti-Rabbit IgG (H + L) (31210 Thermo Pierce, dilution 1:5000) secondary antibody for 1 h at room temperature, and rinsed three times with T-TBS. Protein bands were detected using the Enhanced Chemiluminescence Detection Kit (PE0010, Beijing Sun Technology Co., Beijing, China) and imaged using the ChemiDoc MP Imaging System (Bio-Rad, Hercules, CA, USA). All protein expression levels were normalized using β-actin as a reference.

### 4.8. Establishment of Xenograft Tumor Model in BALB/c Nude Mice

For our study, 5–6-week-old BALB/c nude mice (male) were obtained from Shanghai Slack Laboratory Animal Co., Ltd., Shanghai, China. All animals were maintained at a constant temperature of 25 ± 1 °C with free access to food and drinking water. All animal experimental procedures were performed in accordance with the guidelines and protocols of the Animal Experimental Ethics Committee of Zhejiang University (Zhejiang University, 25468, 20230316 Date of approval 16 March 2023). A subcutaneous tumor model of lung cancer was established to examine the in vivo efficacy of WE-AER against lung cancer. Specifically, 100 μL of 1 × 10^8^ A549 cell suspension was slowly injected into the right axilla of nude mice. Mice were orally treated with 100 μL of saline, PTX (10 mg/kg), or WE-AER (600 mg/kg) once the tumor volume reached 50–100 mm^3^. Every two days, the tumor volume and body weight of the mice were assessed. On day 21 post-treatment, tumor tissues were harvested, and tumor volume was calculated as width^2^ × length × 0.5.

### 4.9. Immunofluorescence Tissue Staining

Tumor sections were embedded in paraffin, followed by dewaxing, antigen repair, permeabilization, and blocking with goat serum. For immunofluorescence, primary antibodies against VEGF and CD31 were applied. The corresponding signals were detected using 488- and Cy3-conjugated secondary antibodies. TUNEL assays were performed using the In Situ Cell Death Detection Kit according to the manufacturer’s protocol. Cell nuclei were stained with DAPI. Sample images were captured using a microscope. Fluorescence intensity was quantified by capturing images from three randomly selected areas, and all images were processed and analyzed using ImageJ software (version 1.51).

### 4.10. H&E Staining

Following dewaxing and washing, the tissue sections were stained with hematoxylin (Hangzhou Seville Co., Ltd., Hangzhou, China) for 5 min. They were then treated with a hydrochloric acid solution for 2 s, differentiated in an ammonia solution (Sinopharm Chemical Reagent Co., Ltd., Shanghai, China) for 15–30 s, and rinsed. The sections were dehydrated with 95% alcohol, counterstained with eosin (Hangzhou Seville Co., Ltd.) for 5–8 s, further dehydrated, mounted, and examined under a microscope.

### 4.11. Cytokine Level Analysis

Blood was collected from mice and centrifuged at 3500 rpm for 15 min to separate the serum. The serum was then analyzed for IL-1β, IL-6, IL-10, and TNF-α levels using ELISA kits, following the manufacturer’s recommendations.

### 4.12. Statistical Analysis

Data are presented as mean ± standard deviation (except for animal experiments and cell viability assays, which require six replicates per sample group, all other experiments require three replicates). All statistical analyses were conducted using GraphPad Prism version 10.1.0 (GraphPad Software Inc., San Diego, CA, USA). Significance was determined using one-way ANOVA with Tukey’s multiple comparison test, with *p* < 0.05 considered statistically significant.

## 5. Conclusions

In summary, WE-AER, an extract derived from AER, has demonstrated significant efficacy in inhibiting the proliferation of A549 cells both in vitro and in vivo. This study also uncovered the underlying molecular mechanism, which involved the induction of apoptosis through the mediation of TGF-β/Foxo/mTOR pathways. Additionally, WE-AER was found to delay the growth of A549 xenografts in vivo by decreasing the expression of VEGF and CD31. It exhibited anti-inflammatory properties, modulated the immune response, and improved the tumor microenvironment. These findings provided a novel perspective on the antitumor mechanism of WE-AER, highlighting its potential as a promising candidate for the treatment of NSCLC.

## Figures and Tables

**Figure 1 ijms-26-08957-f001:**
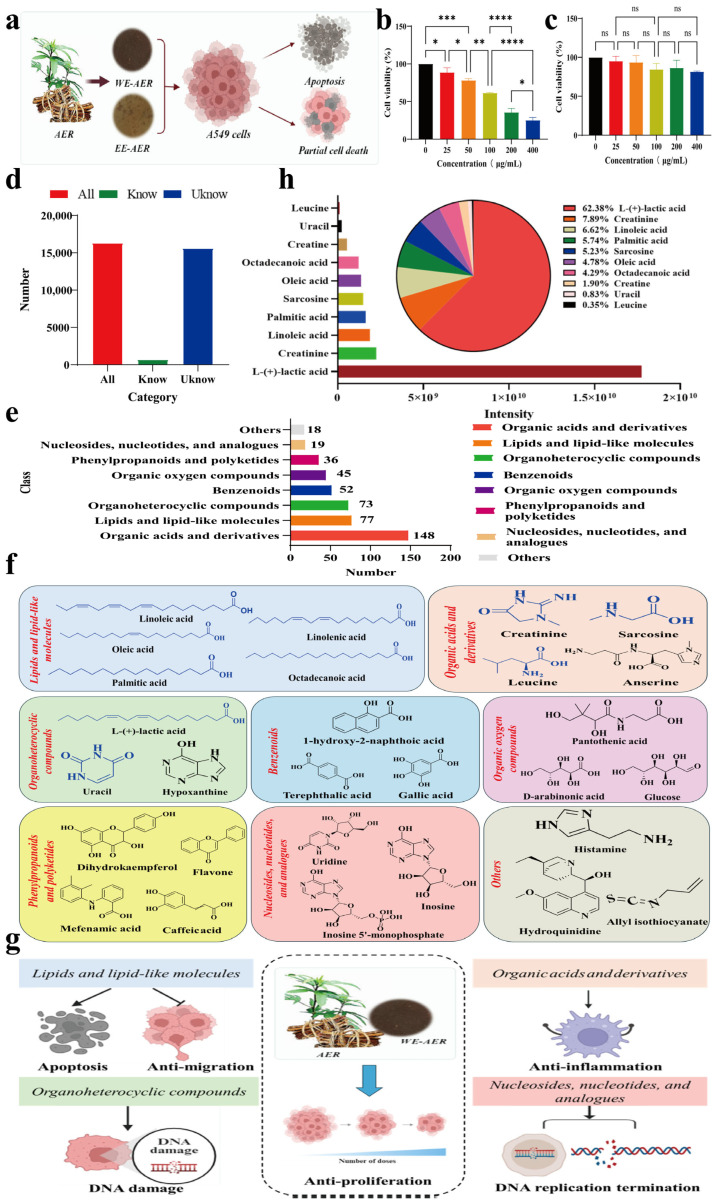
Anti-proliferation effects of tested extracts of *Actinidia eriantha Benth*. root on A549 cells and small molecule compositional analysis. (**a**) Schematic diagram of WE-AER and EE-AER extraction and anticancer efficacy. (**b**,**c**) Relative cell viability of A549 was determined after treatment with WE-AER and EE-AER for 24 h using CCK-8 assay. (**d**) Number of WE-AER small molecules in different categories. (**e**) Quantitative distribution of small-molecule constituents in WE-AER across different categories. (**f**) Representative small molecules and their structural formulas by category in WE-AER (top ten bioactive compounds highlighted in blue). (**g**) Identification of bioactive small molecules in WE-AER with potential anticancer properties and illustration of their mechanistic pathways. (**h**) Qualitative and quantitative analysis of the composition of the top ten small molecules of the WE-AER. ns *p* > 0.05, * *p* < 0.05, ** *p* < 0.01, *** *p* < 0.001, **** *p* < 0.0001. WE-AER: aqueous extract of *Actinidia eriantha Benth*. root; EE-AER: ethanol extract of *Actinidia eriantha Benth*. root.

**Figure 2 ijms-26-08957-f002:**
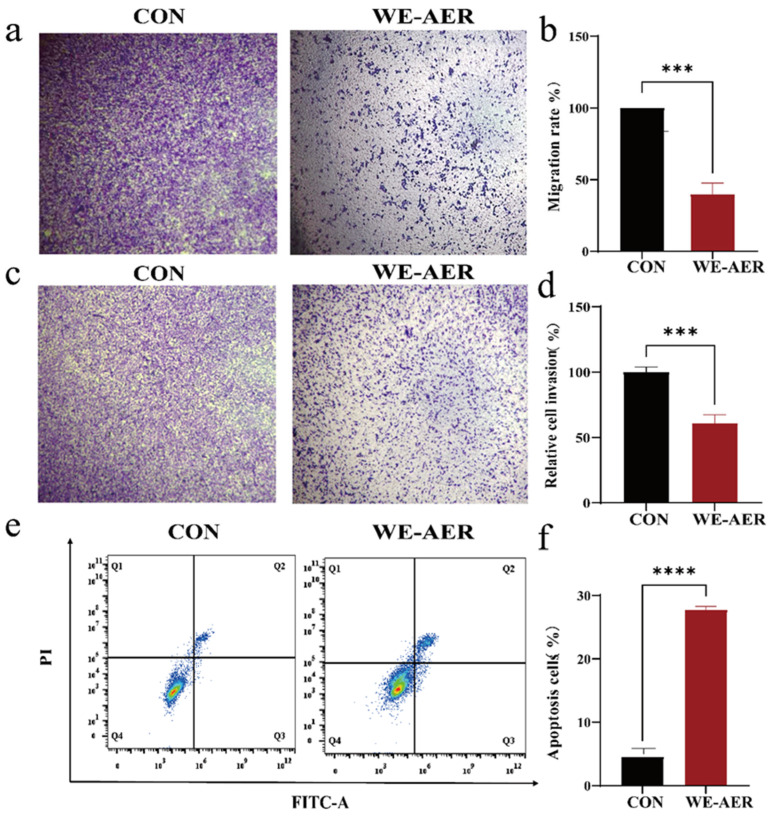
Effects of WE-AER on A549 cell migration, invasion, and apoptosis. (**a**,**b**) Migration and quantitative characterization at 24 h. (**c**,**d**) Invasion and quantitative characterization at 24 h. (**e**,**f**) Apoptosis and quantitative characterization at 24 h. Scale bar: 100 μm. **** *p* < 0.0001, *** *p* < 0.001. WE-AER: aqueous extract of *Actinidia eriantha Benth*. root.

**Figure 3 ijms-26-08957-f003:**
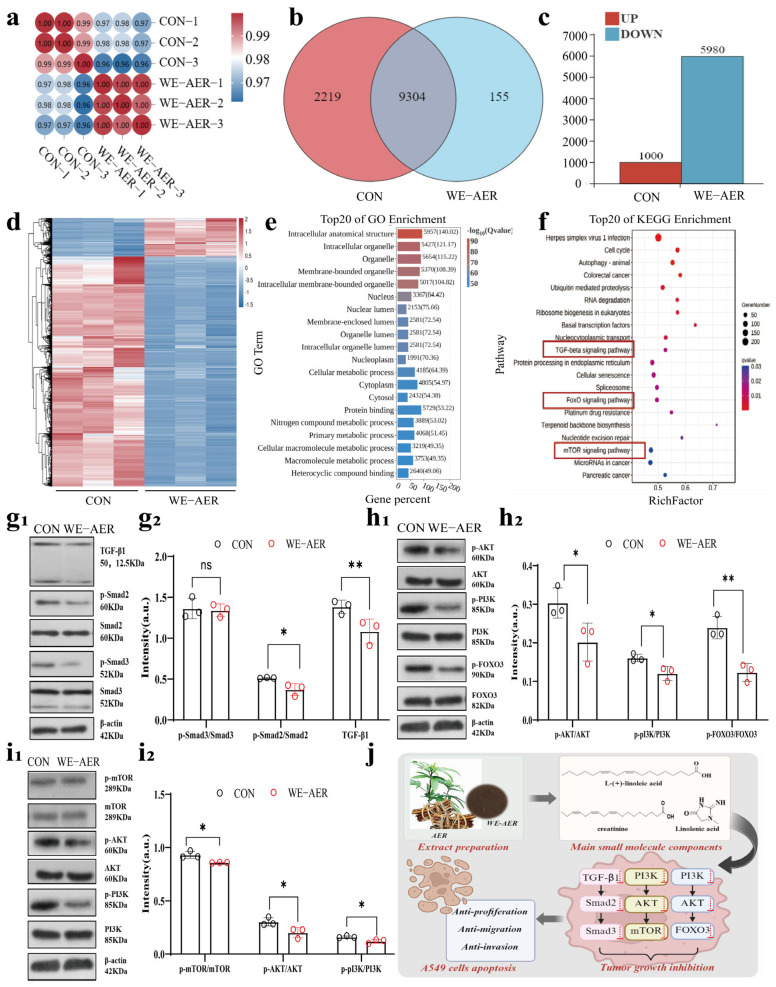
Analysis and validation of the anticancer mechanism of WE-AER. (**a**) Pearson correlation coefficient heatmap between samples in the control (CON) group and the WE-AER intervention group, indicating high intra-group reproducibility and distinct inter-group differences. (**b**) Venn diagram illustrating the number of genes commonly and uniquely expressed in the CON and WE-AER groups. (**c**) The identified up-regulated and down-regulated differentially expressed gene. (**d**) Differential expression heatmap after WE-AER intervention in 4T1 cells. (**e**,**f**) GO and IKEGG pathway enrichment analysis after WE-AER intervention in A549 cells. (**g1**–**i2**) Qualitative and quantitative analysis of the expression levels of proteins involved in TGF-β, FOXO, and mTOR signaling pathways with or without WE-AER treatment. (**j**) Schematic diagram of the antitumor mechanisms of WE-AER. ns *p* > 0.05, * *p* < 0.05, ** *p* < 0.01. PTX: paclitaxel; WE-AER: aqueous extract of *Actinidia eriantha Benth*. root.

**Figure 4 ijms-26-08957-f004:**
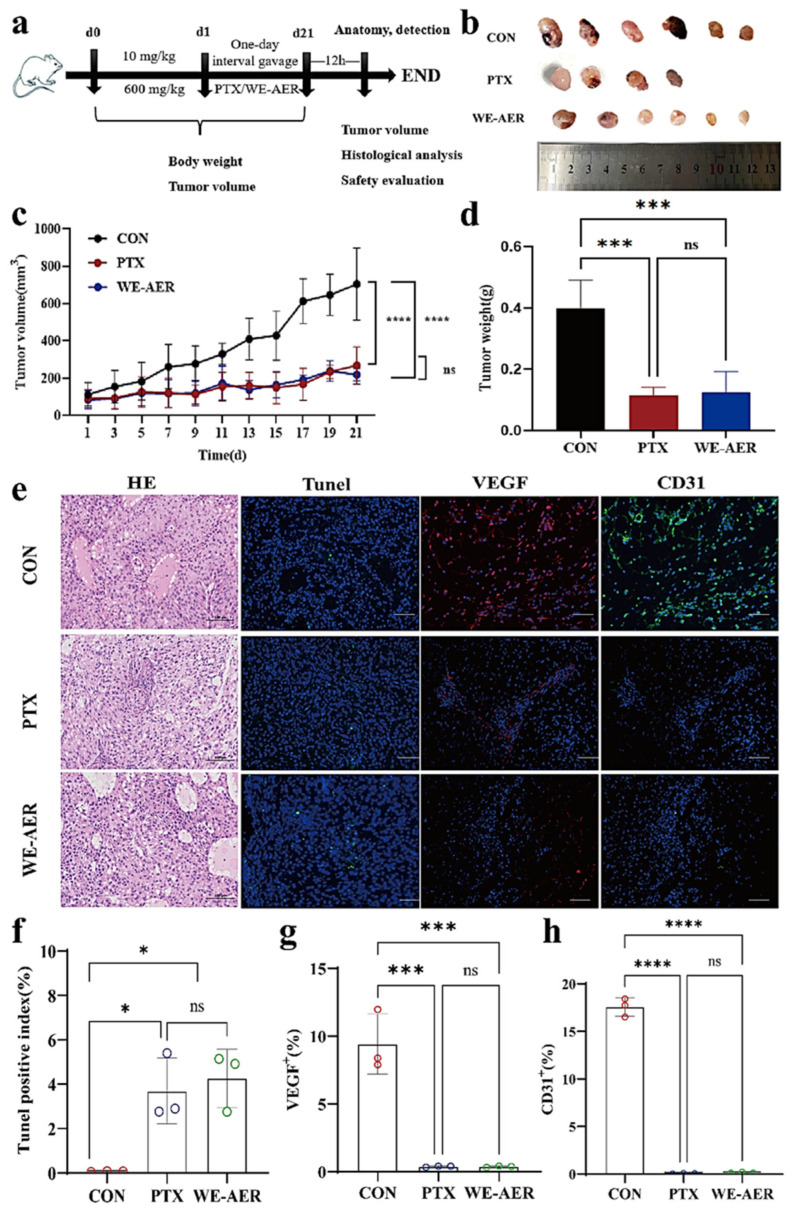
In vivo antitumor efficacy of WE-AER. (**a**) Establishment of A549 mouse model and treatment protocol. (**b**,**c**) Changes in tumor volume after 21 d. (**d**) Tumor weight on day 21. (**e**) Histological analysis and quantification (**f**–**h**) by TUNEL assay, and VEGF and CD31 staining. H&E scale: 200×; immunofluorescence scale: 200×. **** *p* < 0.0001, *** *p* < 0.001, * *p* < 0.05.

**Figure 5 ijms-26-08957-f005:**
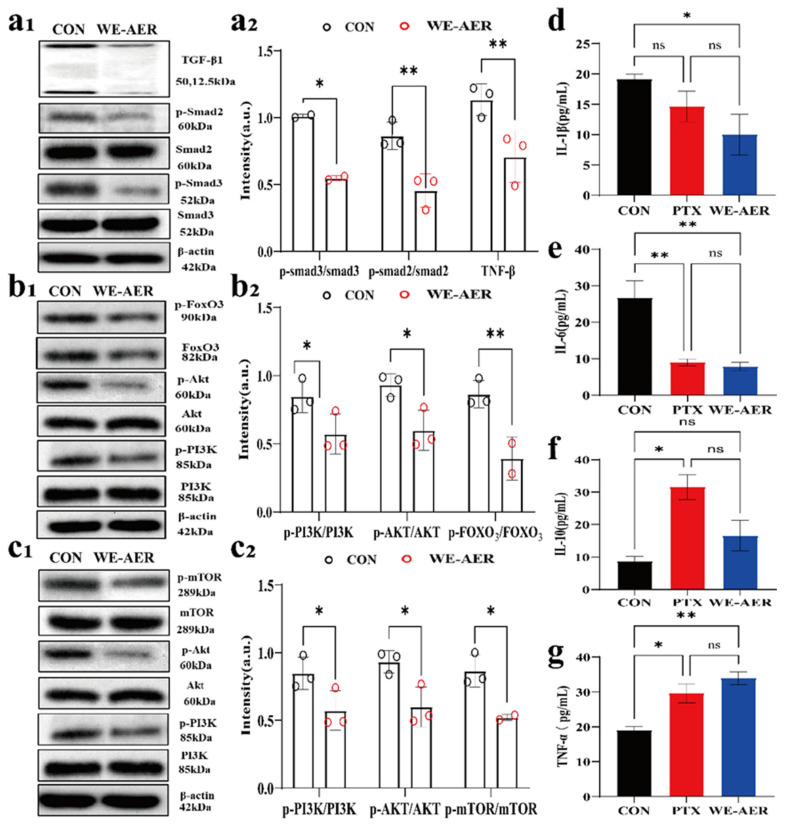
The antitumor mechanism of WE-AER in vivo. Expression levels (**a1**,**b1**,**c1**) and quantification (**a2**,**b2**,**c2**) of TGF-β, FOXO, and mTOR signaling pathways in tumor tissues. (**d**–**g**) Effects of cytokines in serum of treatment groups. ns *p* > 0.05, * *p* < 0.05, ** *p* < 0.01. PTX: paclitaxel; WE-AER: aqueous extract of *Actinidia eriantha Benth*. root.

**Figure 6 ijms-26-08957-f006:**
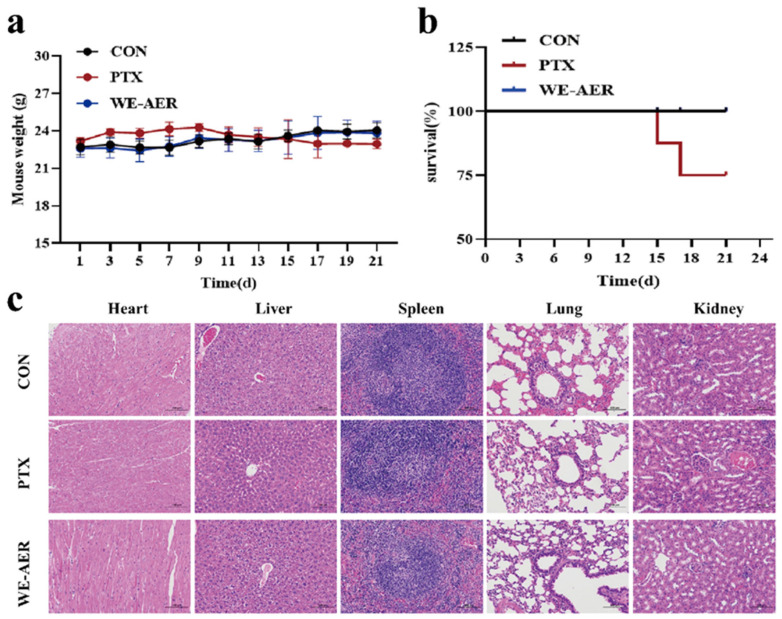
Biological safety evaluation of WE-AER in vivo. (**a**,**b**) Changes in body weight and survival of mice during 21 d of gavage treatment with double-distilled aqueous PTX or WE-AER. (**c**) Histological analysis of sections of heart, liver, spleen, lungs, and kidneys of the double-distilled aqueous PTX or WE-AER treatment groups. H&E scale: 100 μm. PTX: paclitaxel; WE-AER: aqueous extract of *Actinidia eriantha Benth*. root.

**Figure 7 ijms-26-08957-f007:**
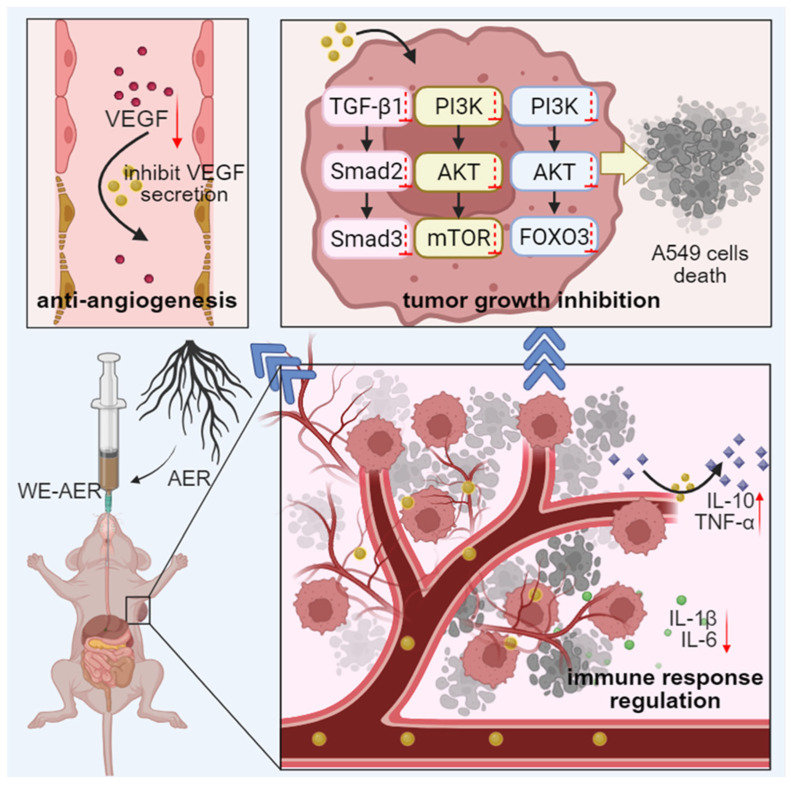
Molecular mechanisms of WE-AER in treating NSCLC tumor growth, angiogenesis, and immune activation.

**Table 1 ijms-26-08957-t001:** Organ index of mice in each group.

Groups	Cardiac Index (g/g)	Liver Index (g/g)	Splenic Index (g/g)	Lung Index (g/g)	Renal Index (g/g)
CON	0.006 ± 0.001	0.067 ± 0.006	0.006 ± 0.001	0.007 ± 0.001	0.016 ± 0.002
PTX	0.005 ± 0.000	0.042 ± 0.003 ****	0.004 ± 0.003	0.007 ± 0.001	0.013 ± 0.001 *
WE-AER	0.006 ± 0.000	0.065 ± 0.002	0.005 ± 0.001	0.006 ± 0.001	0.016 ± 0.001

* *p* < 0.05, **** *p* < 0.0001. PTX: paclitaxel; WE-AER: aqueous extract of *Actinidia eriantha Benth*. root.

## Data Availability

Data will be made available on request.

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
