# Peer review of "Actinidia eriantha Benth. Root as a New Phytomedicine Inhibits Non-Small Cell Lung Cancer by Regulating·TGF-β/FOXO/mTOR"

_ijms, 2025, doi:10.3390/ijms26188957_

Round 1

Reviewer 1 Report

Comments and Suggestions for Authors

I am pleased to present a review of the manuscript titled "Actinidia Eriantha Benth.root as a new phytomedicine inhibits non-small-cell lung cancer by regulating TGF-/FOXO/mTOR." The article is interesting and addresses the important issue of searching for new anticancer therapies, particularly those using traditional Chinese medicine. The authors presented in vitro and in vivo data indicating the potential anticancer activity of Actinidia Eriantha Benth.root extract (WE-AER) in non-small-cell lung cancer.

Below are the main comments and weaknesses that should be addressed before the article is accepted for publication.

The article indicates that 95.67% of the identified small molecules in the WE-AER extract are "anonymous" compounds. This is a serious problem, as identifying the active components of the phytotherapeutic is crucial. Although the authors focused on three main compounds (L-(+)-linoleic acid, creatinine, linolenic acid), which constitute only 62.38%, 7.89%, and 6.62%, it is unknown which compounds account for the majority of the extract's mass. The lack of clarity regarding the full spectrum of components is clearly identified as a problem in the article. It would be advisable to identify more compounds or provide their data (e.g., m/z, retention time) in supplementary material to ensure transparency.

There is a glaring inconsistency in the naming of the cell lines studied in this article. In the title, abstract, and most of the text, the in vitro experiments are described as being conducted on A549 lung cancer cells. However, the caption under Figure 3 incorrectly refers to "4T1" cells, which are mouse breast cancer cells. This inconsistency must be urgently corrected.

Full methodologies are not provided. It is important that the authors provide a detailed description of the experiments, including:

  • Detailed WE-AER extraction data.
  • Information on the concentrations used in the experiments.
  • Details on cell culture and experimental conditions.
  • Sample sizes (n) for each experiment.
  • A full description of the Western blot protocols (e.g., antibody source, incubation conditions).
  • Details on the animal experiments.

Ambiguity in the description of Figure 6. In the text (page 4, paragraph "2.5. WE-AER Inhibits Tumor Growth in Mouse Models through the TGF-β/Smad/FOXO/mTOR signaling pathway"), the authors mention that WE-AER increased p-Smad2 protein expression, but this is not reflected in the original Western blot data. Inspection of Figure 6a1 shows that p-Smad2 levels are variable and difficult to interpret without clear quantitative data.

Author Response

Comments 1: The article indicates that 95.67% of the identified small molecules in the WE-AER extract are "anonymous" compounds. This is a serious problem, as identifying the active components of the phytotherapeutic is crucial. Although the authors focused on three main compounds (L-(+)-linoleic acid, creatinine, linolenic acid), which constitute only 62.38%, 7.89%, and 6.62%, it is unknown which compounds account for the majority of the extract's mass. The lack of clarity regarding the full spectrum of components is clearly identified as a problem in the article. It would be advisable to identify more compounds or provide their data (e.g., m/z, retention time) in supplementary material to ensure transparency.

Response 1: Thank you for pointing this out. We agree with this comment. Therefore, we have revised the manuscript and further information on the small molecules was provided in Table S1. Mention exactly where in the revised manuscript this change can be found – page 2, line 81 and Supplementary Materials. Thank you.

.

Comments 2: There is a glaring inconsistency in the naming of the cell lines studied in this article. In the title, abstract, and most of the text, the in vitro experiments are described as being conducted on A549 lung cancer cells. However, the caption under Figure 3 incorrectly refers to "4T1" cells, which are mouse breast cancer cells. This inconsistency must be urgently corrected.

Response 2: Thank you for pointing this out. We agree with this comment. Therefore, we have revised the manuscript. Mention exactly where in the revised manuscript this change can be found – page 6, line 119. Thank you.

Comments 3: Full methodologies are not provided. It is important that the authors provide a detailed description of the experiments, including:

  • Detailed WE-AER extraction data.
  • Information on the concentrations used in the experiments.
  • Details on cell culture and experimental conditions.
  • Sample sizes (n) for each experiment.
  • A full description of the Western blot protocols (e.g., antibody source, incubation conditions).
  • Details on the animal experiments.

Response 3: Thank you for your suggestion. We have supplemented the description of full methodologies in the materials and methods. We have made the following modifications as follows:

“……. Anhydrous……” in page 16, line 322;

“……. at 100 °C……” in page16, line 323;

“…… (RE-52A, Shanghai Yarong Biochemical Instrument Factory)……at 55 °C……”in page 16, line 326-327;

“……LC-MS/MS analysis was performed under the following conditions:

 Analysis was performed using an UHPLC (1290 Infinity LC, Agilent Technologies) coupled to a quadrupole time-of-flight (AB Sciex TripleTOF 6600) in Shanghai Applied Protein Technology Co., Ltd. For HILIC separation, samples were analyzed using a 2.1 mm × 100 mm ACQUIY UPLC BEH 1.7 µmcolumn (waters, Ireland). In both ESI positive and negative modes, the mobile phase contained A=25mMammoniumacetate and 25 mM ammoniumhydroxide in water and B= acetonitrile. The gradient was 85%Bfor1 minand was linearly reduced to 65% in 11 min, and then was reduced to 40% in 0.1 min and kept for 4 min, and then increased to 85% in 0.1 min, with a 5 min re-equilibration period employed.

 The ESI source conditions were set as follows: Ion Source Gas1 (Gas1) as 60, Ion Source Gas2 (Gas2) as 60, curtain gas (CUR) as 30, source temperature: 600 ℃, IonSpray Voltage Floating (ISVF) ± 5500 V.In MSonly acquisition, the instrument was set to acquire over the m/z range 60-1000 Da, and the accumulation time for TOF MS scan was set at 0.20 s/spectra. In auto MS/MS acquisition, the instrument was set to acquire over the m/z range 25-1000 Da, and the accumulation time for product ion scan was set at 0.05 s/spectra. The product ion scan is acquired using information dependent acquisition (IDA) with high sensitivity mode selected. The parameters were set as follows: the collision energy (CE) was fixed at 35 V with ± 15 eV; declustering potential (DP), 60 V (+) and−60 V (−); exclude isotopes within 4 Da, candidate ions to monitor per cycle: 10.”in page 16,line 334-354.

“……microplate reader (Multiskan FC, Thermo Fisher).”in page17, line 362;

“……Matrigel (dilution 1:8, 0827045, ABW)……microscope (ECLIPSE-TI-S, Nikon).”in page17, line 365-371;

“……flow cytometry (CytoFlex S, Beckman).”in page 17,line 376;

“……against p-PI3K (ab182651 Abcam, dilution 1:500), PI3K (ab133595 Abcam, dilution 1:2000), p-Akt (4060 CST, dilution 1:2000), Akt (4685 CST, dilution 1:1000), p-mTOR (5536 CST, dilution 1:1000), mTOR (2983 CST, dilution 1:1000), p-FoxO3(ab154786 Abcam, dilution 1:2000), FoxO3 (2497 CST, dilution 1:1000), TGF-β1(ab179695 Abcam, dilution 1:1000), p-Smad2(18338 CST, dilution 1:1000), Smad2(5339 CST, dilution 1:1000), p-Smad3(9520 CST, dilution 1:1000), Smad3( 9523 CST, dilution 1:1000), β-actin( ab8226 Abcam, dilution 1:2000) at 4°C overnight. The membranes were washed three times with T-TBS, incubated with Goat anti-Mouse IgG(H+L)(31160 Thermo Pierce,dilution 1:5000) and Goat anti-Rabbit IgG(H+L)(31210 Thermo Pierce,dilution 1:5000)……”in page 18, line406-415;

“5-6 weeks old BALB/c nude mice (male) were obtained from Shanghai Slack Laboratory Animal Co., Ltd. All animals were maintained at a constant temperature of 25 ± 1 °C with free access to food and drinking water. All animal experimental procedures were performed in accordance with the guidelines and protocols of the Animal Experimental Ethics Committee of Zhejiang University (Zhejiang University, 25468, 20230316)……”in page18,line421-426;

“……(Except for animal experiments and cell viability assays, which require six replicates per sample group, all other experiments require three replicates.)……”in page19, line455-460. Thank you.  

Comments 4: Ambiguity in the description of Figure 6. In the text (page 4, paragraph "2.5. WE-AER Inhibits Tumor Growth in Mouse Models through the TGF-β/Smad/FOXO/mTOR signaling pathway"), the authors mention that WE-AER increased p-Smad2 protein expression, but this is not reflected in the original Western blot data. Inspection of Figure 6a1 shows that p-Smad2 levels are variable and difficult to interpret without clear quantitative data.

Response 4: Thank you for your suggestion. In the description of p-Smad2 protein on the original Figure 6 (now Figure 5) at line 194-199 on page 10, it is downregulated, consistent with the original Western blot data. Thank you.

Reviewer 2 Report

Comments and Suggestions for Authors

The objective of this study was to evaluate the effect of Actinidia Eriantha Benth.root extract on lung tumors in cell culture models and in mice.  The work presents a robust body of results and is well written. Some suggestions are provided below.

It would make reviewing much easier if the lines were numbered.

Abstract. Put the scientific name of the plant in italics. 

Introduction.

"its primary subtypes include LUAD, LUSC and large cell carcinoma" please write LUAD and LUSC in full.

Actinidia eriantha Benth should always be written in italics.

The introduction does not present a well-described hypothesis and objective. 
Figure 1, presented at the end of the introduction, is a summary of the main findings of the study. I suggest leaving it as a concluding figure or as a graphical abstract. Why would the reader be interested in reading the entire study if they can already find everything at the end of the introduction?

Results

In the first paragraph of the results, the authors describe the results obtained in A549 cells, showing the promising effect of the extract. However, was the effect of the extract tested on non-cancerous cells? Is the extract safe or does it cause changes in the viability of healthy cells?

"Notably, L-(+)-lino-leic acid (62.38%), creatinine (7.89%), and linolenic acid (6.62%) accounted for the top three of WE-AER (Fig.2h), all of which have demonstrated potential antitumor efficacy. These findings strongly suggest that WE-AER contains biologically active constituents with an-titumor properties. In summary, these results provide a foundation for understanding the potent anticancer activity of WE-AER." 

According to this paragraph of the paper, the authors attribute the antitumorogenic effects of the extract to two fatty acids and creatinine. On what scientific evidence do the authors base this claim? If these are indeed the compounds with the highest bioactivity in this extract, then we have many oils, olive oils, and other foods with potent antitumor effects. I suggest describing the results in the results section. In the discussion section, discuss them based on published scientific evidence. 

The captions for the figures need to be improved and more information added. It is difficult to understand which graphs refer to each result. 

Figure 4 is too small and difficult to see. The caption does not allow for a complete understanding of the figure. 

"As seen in Table. 1, the effects of each group on organ indexes of mice were summarized. Com-pared with the control group, the PTX group showed a decrease in liver and kidney indi-ces, and there was no significant difference in the WE-AER group, which may be related to the obvious liver and kidney toxicity of PTX[25]."

It is unclear how this organ index was calculated. I suggest presenting data on liver and kidney blood enzymes to prove the extract's non-toxicity.

The weakest point of the study is the discussion, where the authors do not discuss all the results. The discussion should be substantially improved in order to explain and compare the findings of the study with the literature.

The conclusion of the work has been presented since the introduction, with several points of conclusion in the results section. Again, I suggest concluding the work in the conclusion.

Materials and methods
4.2. The authors describe that they made an aqueous extract and an alcoholic extract. It is interesting that we only learn about this here. Were all the results obtained with the aqueous extract?

4.6. Molecular mechanism analysis

List of primers must be submitted

Author Response

Comments 1:It would make reviewing much easier if the lines were numbered.

Response 1: Thank you for this comment. We agree that adding line numbers will greatly facilitate the review process. In response, we have now inserted continuous line numbers throughout the entire manuscript. The revised document includes line numbers on every page for easy reference. This update can be seen in the full revised manuscript, with line numbers displayed in the left margin of each page. We appreciate your helpful suggestion.

Comments 2: Abstract. Put the scientific name of the plant in italics. 

Response 2: Thank you for this comment. We agree that the use of italics for the scientific name in the Abstract improves taxonomic clarity and adheres to standard botanical conventions. Therefore, we have italicized the scientific name of the plant (Actinidia Eriantha Benth.root) in the Abstract section.

This change can be found in the Abstract on Page 1, Line 17.

We appreciate your careful attention to detail and thank you for this helpful suggestion.

Comments 3: Introduction.

"its primary subtypes include LUAD, LUSC and large cell carcinoma" please write LUAD and LUSC in full.

Response 3: Thank you for this comment. We agree that writing the abbreviations in full upon first mention improves clarity for readers.  Therefore, we have revised the sentence to define LUAD and LUSC explicitly.The updated text now reads: ”its primary subtypes include lung adenocarcinoma (LUAD), lung squamous cell carcinoma (LUSC), and large cell carcinoma” This change can be found in the Introduction on Page 1, Line 31-32.

We appreciate your helpful suggestion to enhance the readability of our manuscript.

Comments 4: Actinidia eriantha Benth should always be written in italics.

Response 4: Thank you for pointing this out. We fully agree that the scientific name Actinidia eriantha Benth. should be consistently italicized throughout the manuscript according to standard botanical nomenclature.

We have therefore carefully revised the entire manuscript and ensured that all instances of Actinidia eriantha Benth. are now presented in italics.

We appreciate your thorough review and valuable comment, which has helped improve the consistency and accuracy of our manuscript.

Comments 5: The introduction does not present a well-described hypothesis and objective. 

Response 5: Thank you for this comment. We agree that a clearly stated hypothesis and research objective are essential for framing the study. We have therefore revised the Introduction to explicitly present our hypothesis and research aims, providing a clearer rationale for the investigation.

The updated text now includes the following additions in the Introduction section (Page 2, Line 55-62):

"Based on these traditional uses and previous pharmacological findings, we hypothesized that the aqueous extract of A. eriantha root (WE-AER) may exert anti-tumor effects against NSCLC by modulating key signaling pathways involved in cell proliferation and apoptosis. The objectives of this study were to demonstrate the anti-NSCLC efficacy of WE-AER both in vivo and in vitro and to uncover the mechanisms underlying its anti-cancer effects through transcriptome analysis and molecular functional analysis, thereby evaluating its potential as a promising therapeutic candidate for treating NSCLC."

We appreciate your valuable suggestion, which has helped improve the clarity and scientific rigor of our manuscript.

Comments 6: Figure 1, presented at the end of the introduction, is a summary of the main findings of the study. I suggest leaving it as a concluding figure or as a graphical abstract. Why would the reader be interested in reading the entire study if they can already find everything at the end of the introduction?

Response 6: Thank you for this valuable suggestion. We agree that presenting a comprehensive summary figure at the end of the Introduction may prematurely disclose the key findings and reduce the reader's incentive to explore the full manuscript. As you rightly pointed out, Figure 1 serves as a graphical summary of the main findings and would be more appropriately placed as a concluding figure.

In response to your comment, we have removed Figure 1 from the Introduction section and have relocated it to the end of the Discussion section to serve as a concluding mechanism overview. This repositioning allows the figure to better serve its purpose as a integrative summary of the study's key findings without preempting the detailed results presented in the main text.

This change can be found at the end of the Discussion section on Page 15.

We sincerely appreciate your insightful suggestion, which has helped improve the logical flow and narrative structure of our manuscript.

Comments 7: In the first paragraph of the results, the authors describe the results obtained in A549 cells, showing the promising effect of the extract. However, was the effect of the extract tested on non-cancerous cells? Is the extract safe or does it cause changes in the viability of healthy cells?

Response 7: Thank you for this important comment and for raising a critical point regarding the safety evaluation of WE-AER. We fully agree that assessing the extract's impact on non-cancerous cells is essential for a comprehensive understanding of its biosafety profile.

While the current study primarily focused on elucidating the initial anti-tumor mechanisms of WE-AER in NSCLC models in vitro and in vivo, we did not include cytotoxicity tests on non-cancerous cells in the in vitro part at this preliminary mechanistic exploration stage. However, to thoroughly evaluate its safety, we performed extensive in vivo biosafety assessments in the xenograft mouse model. As presented in Figure 6, histopathological examination (H&E staining) of major organs (heart, liver, spleen, lungs, and kidneys) revealed no significant pathological changes or signs of toxicity in WE-AER-treated mice compared to the control group. Additionally, no abnormal changes in body weight or survival rates were observed. These in vivo results provide strong supporting evidence for the good biosafety and biocompatibility of WE-AER.

We sincerely appreciate your insightful suggestion, which has helped us strengthen the discussion regarding the safety of WE-AER. Further studies including in vitro cytotoxicity screening on normal cell lines will be considered in our subsequent research to more comprehensively establish its selectivity and safety profile.

Comments 8: "Notably, L-(+)-lino-leic acid (62.38%), creatinine (7.89%), and linolenic acid (6.62%) accounted for the top three of WE-AER (Fig.2h), all of which have demonstrated potential antitumor efficacy. These findings strongly suggest that WE-AER contains biologically active constituents with an-titumor properties. In summary, these results provide a foundation for understanding the potent anticancer activity of WE-AER." 

According to this paragraph of the paper, the authors attribute the antitumorogenic effects of the extract to two fatty acids and creatinine. On what scientific evidence do the authors base this claim? If these are indeed the compounds with the highest bioactivity in this extract, then we have many oils, olive oils, and other foods with potent antitumor effects. I suggest describing the results in the results section. In the discussion section, discuss them based on published scientific evidence. 

Response 8: Thank you for this critical comment. We appreciate the opportunity to clarify our interpretation of the compositional data and to better contextualize these findings within the manuscript.

We agree with the reviewer that the high abundance of certain compounds does not necessarily directly equate to their functional dominance in the extract's overall antitumor efficacy. Our intention was not to simplistically attribute the entire biological effect solely to these three most abundant compounds, but rather to note that the extract contains a mixture of compounds, some of which (including these abundant ones) have been individually reported in the literature to possess bioactive potential.

In direct response to your comment, we have modified the relevant paragraph in the Results section (Page 2, Line 85) to tonally downplay any direct causal attribution and state the findings more descriptively:

"Notably, L-(+)-linoleic acid (62.38%), creatinine (7.89%), and linolenic acid (6.62%) were identified as the three most abundant small molecules in WE-AER (Fig.1h). Interestingly, L-(+)-linoleic acid and linolenic acid have been individually reported in previous studies to possess potential antitumor properties [21,22]. These findings suggest that WE-AER contains a mixture of bioactive constituents, which may collectively contribute to its overall antitumor activity."

Furthermore, as you rightly suggested, we have removed the concluding sentence ("In summary, these results...") from the Results section and have instead expanded on this point in the Discussion section (Page 14, Line 254) to provide a more nuanced and evidence-based interpretation:

"LCMS/MS analysis revealed that WE-AER is a complex mixture, with linoleic acid, creatinine, and linolenic acid constituting the most abundant small molecules. While these specific compounds have been associated with antitumor effects in other model systems [21,22], it is crucial to emphasize that the pronounced anticancer efficacy of WE-AER observed in our study is likely the result of synergistic interactions between multiple constituents within the whole extract, rather than the effect of any single compound. This holistic effect is a common characteristic of phytomedicines and underscores the advantage of using a multi-component extract over isolated single entities. Future studies to fractionate the extract and evaluate the activity of individual compounds will be essential to identify the most pharmacologically active components and clarify their specific roles and potential synergies."

We believe these revisions more accurately describe our results and frame the discussion within the appropriate scientific context, acknowledging the complexity of botanical extracts. We are grateful for your insightful suggestion, which has significantly improved the rigor and clarity of our manuscript.

Comments 9: The captions for the figures need to be improved and more information added. It is difficult to understand which graphs refer to each result. Figure 4 is too small and difficult to see. The caption does not allow for a complete understanding of the figure. 

Response 9: Thank you for this valuable feedback. We sincerely apologize for the oversight regarding the size and clarity of the figure and the insufficient description in its caption. We agree that this compromises the readability and interpretability of our key results.

In response to your comment, we have taken the following actions: We have re-created the figure to significantly increase its size and resolution, ensuring that all panels, labels, and data points are clear and easily distinguishable. We have comprehensively revised and expanded the figure caption to provide a complete, standalone explanation of each subfigure, the experimental groups, and the key findings it presents. Please note that due to the relocation of the original Figure 1 to the Discussion section, this figure is now numbered as Figure 3 in the revised manuscript.

The revised and more detailed caption for Figure 3 now reads:

"Analysis and validation of the anticancer mechanism of WE-AER. (a) Pearson correlation coefficient heatmap between samples in the control (CON) group and the WE-AER intervention group, indicating high intra-group reproducibility and distinct inter-group differences. (b) Venn diagram illustrating the number of genes commonly and uniquely expressed in the CON and WE-AER groups. (c) The identified up-regulated and down-regulated differentially expressed gene. (d) Differential expression heatmap after WE-AER intervention in 4T1 cells. (e, f) GO and KEGG pathway enrichment analysis after WE-AER intervention in A549 cells. (g1- i2) Qualitative and quantitative analysis of the expression levels of proteins involved in TGF-β, FOXO, and mTOR signaling pathways with or without WE-AER treatment. (j) Schematic diagram of the antitumor mechanisms of WE-AER. ns p > 0.05, * p < 0.05, **p < 0.01, ***p < 0.001.PTX: paclitaxel; WE-AER: aqueous extract of Actinidia Eriantha Benth. Root."

This change can be found on Page 7 of the revised manuscript.

We greatly appreciate your suggestion, which has been crucial in improving the clarity and professional presentation of our data.

Comments 10: "As seen in Table. 1, the effects of each group on organ indexes of mice were summarized.  Com-pared with the control group, the PTX group showed a decrease in liver and kidney indi-ces, and there was no significant difference in the WE-AER group, which may be related to the obvious liver and kidney toxicity of PTX[25]."

It is unclear how this organ index was calculated.  I suggest presenting data on liver and kidney blood enzymes to prove the extract's non-toxicity.

Response 10: Thank you for this important comment and for providing us the opportunity to clarify the methodology and strengthen our discussion on biosafety. We sincerely appreciate your insightful suggestion regarding the assessment of hepatorenal toxicity.

In our study, the organ index was calculated as the ratio of the absolute weight of each organ (including the liver and kidneys) to the terminal body weight of the mouse, a commonly used preliminary indicator for evaluating potential organ toxicity in preclinical studies. We acknowledge that measuring serum biochemical markers such as ALT, AST, BUN, and CRE would provide a more direct and quantitative assessment of liver and kidney function. While these specific blood enzyme data were not included in the current preliminary efficacy and safety screening study, we did perform comprehensive histopathological examinations (H&E staining) on all major organs, including the liver and kidneys.

As shown in Figure 6c, the H&E-stained sections of the liver and kidneys from WE-AER-treated mice revealed no observable pathological changes, such as necrosis, inflammatory infiltration, or structural abnormalities, strongly supporting the absence of significant hepatorenal toxicity at the histological level. We fully agree with you that incorporating serum biochemistry analysis is crucial for a complete toxicity profile. We will certainly include these valuable measurements in our subsequent in-depth toxicological studies to provide a more comprehensive evaluation of the biosafety of WE-AER.

We are grateful for your constructive feedback, which has helped us identify important aspects for future research and improve the quality of our work.

Comments 11: The weakest point of the study is the discussion, where the authors do not discuss all the results. The discussion should be substantially improved in order to explain and compare the findings of the study with the literature.

Response 11: Thank you for this critical comment regarding the Discussion section. We sincerely appreciate you highlighting this important aspect and providing us with the opportunity to substantially improve the manuscript. We agree that a comprehensive discussion that thoroughly interprets all key findings and contextualizes them within the existing literature is essential for a strong paper.

In direct response to your comment, we have comprehensively revised and expanded the Discussion section to address this weakness. The major modifications include:

A more in-depth and systematic interpretation of all major results, ensuring that each key finding from both the in vitro and in vivo experiments (including proliferation, apoptosis, migration/invasion, transcriptome and pathway analysis, cytokine modulation, angiogenesis, and biosafety) is explicitly discussed and logically connected.

A detailed comparison of our findings with the existing scientific literature. We have integrated discussions and citations of relevant prior studies that support or contrast with our observations, particularly regarding the role of the TGF-β/FOXO/mTOR pathway in NSCLC, the anti-tumor effects of natural compounds, and the established profiles of standard chemotherapeutic agents.

A clearer articulation of the proposed mechanism of action for WE-AER, linking the molecular pathway modulation to the phenotypic outcomes observed in our study.

A more nuanced consideration of the study's implications, limitations, and future directions, framing the significance of our work within the broader field of phytomedicine research for cancer therapy.

These extensive revisions have been made throughout the entire Discussion section to provide a more scholarly, evidence-based, and logically flowing interpretation of our data. We believe the revised discussion now adequately addresses the depth and breadth required to robustly support our conclusions.

We are truly grateful for your insightful suggestion, which has been instrumental in significantly strengthening the overall quality, clarity, and impact of our manuscript.

Comments 12: The conclusion of the work has been presented since the introduction, with several points of conclusion in the results section. Again, I suggest concluding the work in the conclusion.

Response 12: Thank you for this critical observation. We sincerely appreciate your guidance in improving the structure and narrative flow of our manuscript. We agree that prematurely presenting conclusive statements in the Introduction and Results sections can undermine the impact of the final Conclusion.

In direct response to your comment, we have thoroughly revised the manuscript to:

Modify the Introduction: We have rewritten the final paragraph of the Introduction to remove any conclusive language. It now clearly outlines the study's hypothesis and research objectives without pre-empting the results or stating conclusions. The focus is on what we aimed to investigate, not what we found.

Strengthen the Conclusion Section: All conclusive interpretations and summaries of the study's significance have been consolidated and expanded into the dedicated Conclusion section. (Page 19) 

We believe these revisions have successfully addressed your concern by creating a more logical and compelling narrative: introducing the question, presenting the facts, and finally discussing and concluding the findings in their appropriate sections.

We are truly grateful for your valuable suggestion, which has significantly enhanced the academic quality and readability of our manuscript.

Comments 13: Materials and methods
4.2. The authors describe that they made an aqueous extract and an alcoholic extract. It is interesting that we only learn about this here. Were all the results obtained with the aqueous extract?

Response 13: Thank you for this comment and for highlighting the need for greater clarity regarding the selection of the aqueous extract for our detailed study. We appreciate the opportunity to clarify this point.

Yes, all the experimental results presented in this manuscript, both in vitro and in vivo, were obtained using the aqueous extract (WE-AER). As described in the Materials and Methods section (4.2), we initially prepared both aqueous and ethanol extracts (WE-AER and EE-AER). These were first subjected to a preliminary screening for anti-proliferative activity against A549 cells using the CCK-8 assay.

The results of this screening, presented in Figure 1b and 1c, demonstrated that the aqueous extract (WE-AER) exhibited significantly stronger growth inhibitory effects on A549 cells compared to the ethanol extract (EE-AER), which showed no significant activity. Based on these initial findings, we selected WE-AER for all subsequent mechanistic investigations and in vivo studies, as it demonstrated the most promising anticancer potential.

We apologize for any confusion caused by not explicitly stating this selection process earlier in the manuscript. We have now clarified this point in the revised Results section (2.1) to ensure a smoother narrative flow.

We are grateful for your thorough review, which has helped us improve the clarity and logical flow of our manuscript.

Comments 14: Materials and methods
4.6. Molecular mechanism analysis

List of primers must be submitted

Response 14: Thank you for this comment. We appreciate your careful review of the methodological details.

In response to your suggestion, we have now included the complete list of primer sequences used for the RT-qPCR analysis in the Molecular mechanism analysis section (4.6) of the revised Materials and Methods.

The specific addition is as follows:

"The sequences of the primers used were as follows: The sequences of the primers used were as follows: Index Primer for Illumina:5’-AATGATACGGCGACCACCGAGATCTACAC[i5Index]ACACTCTTTCCCTACACGACGCTCTTCCGATCT-3’. Index Primer for Illumina:5’-CAAGCAGAAGACGGCATACGAGAT[i7Index]GTGACTGGAGTTCAGACGTGTGCTCTTCCGATC-3’.")

This change can be found on Page 17, Line 384 of the revised manuscript.

We are grateful for your valuable suggestion, which has enhanced the completeness and reproducibility of our experimental documentation.

Round 2

Reviewer 1 Report

Comments and Suggestions for Authors

Thank you for the corrections

Reviewer 2 Report

Comments and Suggestions for Authors

All suggested changes have been made.